# The Relevance of Chronological and Biological Aging in the Progression of Multiple Sclerosis

**DOI:** 10.3390/healthcare13202619

**Published:** 2025-10-17

**Authors:** Patricia Mulero, Alba Chavarría-Miranda, Nieves Téllez

**Affiliations:** Neurology Department, Hospital Clínico Universitario de Valladolid, 47003 Valladolid, Spain; achavarria@saludcastillayleon.es (A.C.-M.); mntellez@saludcastillayleon.es (N.T.)

**Keywords:** multiple sclerosis, biological age, chronological age, telomere length, epigenetic changes, senescence, aging, senolytics, lifestyle

## Abstract

Chronological age (C-Age), determined by the time elapsed since the birth of an individual, is considered one of the main risk factors for the onset and prognosis of multiple sclerosis (MS). Biological age (B-Age), in contrast, conditioned by genetic, lifestyle, comorbidity, and environmental factors, defines the aging of tissues that contributes to the decline of organ function, the loss of functional reserve, and decrease in the regenerative capacity. In this context immunosenescence is increasingly evidenced as a factor that contributes to the MS progressive course and loss of efficacy of MS drugs. B-Age can be estimated through different measurement strategies such as telomere length, epigenetic clocks and biomarker composites. These biomarkers are gaining attention in MS research since they seem to be associated with disability progression and are modulated by lifestyle interventions. This review summarizes the roles of C-Age and B-Age in MS and highlights implications for prognosis and therapeutic development.

## 1. Introduction

Chronological age (C-Age), determined by the time elapsed since the birth of an individual, is one of the main risk factors for the development of neurodegenerative diseases and their prognosis [1]. However, there is still great heterogeneity in the clinical outcomes despite the classical concept of age [2]. In this context, in recent years the concept of biological age (B-Age) has gained traction. B-Age is a construct that reflects tissue damage, functional reserve, and regenerative capacity [3]. This aspect is determined by genetic factors but is also modifiable by lifestyle habits, which indicates that aging is a “plastic” phenomenon (Figure 1) [4]. According to B-Age, an individual can show “accelerated” or “decelerated” aging in comparison to C-Age, which influences and modulates the risk of developing a disease or the disease trajectory [4].

Multiple sclerosis (MS) is a disease with a broad clinical phenotype, where some patients acquire an early disability and others remain progression free for the rest of their lives. There are well-known radiological and clinical prognostic factors that impact the disease course, and C-Age also seems to be a determinant. However, once again, individuals of the same age show a significant heterogeneity in the disease course despite sharing similar baseline prognostic factors. A hypothetical explanation to at least part of this disparity lies in the individual’s B-Age. Here, we review the relevance of C-Age and the emerging role of B-Age in this neurodegenerative disease.

## 2. Impact of Chronological Age on Multiple Sclerosis

C-Age is a high impact factor for the incidence of MS. The highest incidence rate appears between 20 and 40 years of age, although the disease can onset at any age [5]. Approximately, 2–5% of patients have a pediatric disease (under 17 years) [6] and an estimated 8–10% of patients debut at over 50 years of age, which is called “late onset” [7]. These two populations of MS patients show a significantly different clinical phenotype. In between these two extremes, there is a continuum of intermediate course trajectories.

Pediatric MS patients evolve almost exclusively with a relapsing course, whilst older patients carry a higher risk of developing progressive disease phenotypes with a lower annualized relapse rate [8,9,10]. Epidemiological studies show that most untreated relapsing MS patients progress into a secondary course, where the disease becomes more steadily progressive, with or without relapses. Research data suggest a median time to reach a progressive phase of about 15–20 years after the onset of MS. Also, it is well known that children with relapsing MS onset take longer to reach a progressive phase of disease in comparison to adults, however this occurs at a younger age [8]. This means that patients with a history of pediatric MS accumulate physical and cognitive disabilities at a younger age than patients with adult MS onset [11,12]. In addition, the average age at diagnosis of progressive MS in adults is 10 years older than that of relapsing forms, and the age at onset of progression is highly similar between primary progressive and secondary progressive disease [13,14,15]. On the one hand, older patients experience shorter latency to progression and are more likely to experience incomplete disability recovery following relapses [16]. These observations suggest a strong relationship between C-Age and disability progression. On the other hand, it is well known that some MS patients remain free of disability progression until older ages or are even completely progression free. This suggests that C-Age is a key player but not the only one, and that there are other aspects influencing progressive phenotypes.

## 3. Biological Age in Multiple Sclerosis: What We Know

Aging is a natural process primarily caused by a progressive accumulation of deleterious changes to normal metabolism at a molecular, cellular, and tissue level, exacerbated by environmental toxins and an unhealthy lifestyle. These damages affect tissue functions and tissue capacity for regeneration. The “functionality” of tissues and organs is what we call B-Age. Defining a patient’s biological age may offer more precision in determining the role of the aging process than C-Age does. Scientific evidence supports this statement in different diseases such as breast cancer [17], stroke, and neurodegenerative disease like dementia [18,19].

A recent meta-analysis encompassing 13 cohorts demonstrated that epigenetic alterations in DNA—one of the principal mechanisms driving biological aging (B-Age)—serve as independent predictors of all-cause mortality, irrespective of chronological age (C-Age) and after adjustment for conventional risk factors [20]. Another recent systematic review and meta-analysis have observed that epigenetic alterations were significantly related to mortality of cardiovascular disease, cancer, and diabetes [21].

B-Age can be estimated through different measurement strategies. Biomarkers corresponding to metabolic activity or inflammation correlate with biological functions rather than C-Age and therefore might predict the functional capacity of a tissue or organ. Three main system measures have been proposed: telomere length (TL), epigenetic clocks and biomarker composites (See Table 1). However, no single definitive measure of biological aging currently exists. Several authors have studied the correlation between these different techniques with non-conclusive results [22]. A possible explanation for this discrepancy is that TL, the various proposed epigenetic clocks, and biomarker composites capture distinct molecular pathways of aging, each possessing specific advantages in the investigation of disease-specific mechanisms [23].

### 3.1. Telomere Length

Telomeres are protein structures and nucleotide repeats (TTAGGG) localized at the edge of chromosomes. They shorten with each cell division and are necessary for the stability and protection of genomic DNA [32,33]. Telomeres participate in signaling pathways that control cell proliferation, thereby determining the cellular lifespan. Consequently, they act as biological clocks that define how many times a cell can divide. Hence, they are considered biological clocks that determine the number of divisions that a cell undergoes [34]. In this process, the function of the telomerase enzyme, a ribonucleoprotein that helps in the maintenance of the telomere length, is essential. In most somatic cells, telomerase activity is limited, leading to progressive telomere shortening that ultimately results in replicative senescence; this is an irreversible arrest of the cell cycle, which is a main mechanism underlying aging. The evidence points out that besides C-Age, certain lifestyle factors such as smoking, obesity, physical inactivity, and an unhealthy diet can influence telomere shortening, as can chronic inflammation and oxidative stress, both of which may impair cellular function [35,36].

Several studies support the association between telomere length (TL) and cardiovascular diseases [37], dementia [38], and autoimmune diseases such as lupus or arthritis rheumatoid [39]. The relation between telomere length and MS have also been explored. A systematic review [24] analyzed six studies and found that four of them [25,26,27,40] reported significantly shorter leukocyte telomere lengths, and observed differences compared to healthy controls (*p* = 0.003 in meta-analysis). In this review, shorter telomeres in patients with MS were found to be associated, independently of age, with greater disability, reduced brain volume, higher relapse rates, and a faster conversion from relapsing to progressive MS. This systematic review emphasized that individuals with MS generally exhibit shorter telomeres in blood cells compared to healthy controls, and it has been suggested that TL measurement could serve as a potential biomarker for evaluating and predicting clinical phenotypes of MS [24].

In recent years, various articles have been published attempting to link telomere length with the risk of developing multiple sclerosis through Mendelian randomization analysis. To date, the results are controversial, as two of the studies have shown an association between shorter telomere length and a higher risk of MS [41,42], while a third study describes longer telomere length as being associated with an increased risk [43].

### 3.2. Epigenetic Clocks

Another biomarker of B-Age is related to the critical role that epigenetic changes play in aging. There are a huge number of B-Age measurement systems called “epigenetic clocks” based on methylation patterns in different DNA regions [44]. DNA methylation (DNAm) refers to heritable yet reversible mechanisms of genetic regulation that serve as an interface through which environmental and genetic factors can influence the genome. Several types of epigenetic clocks have been developed to estimate biological age (B-Age) in both healthy individuals and those with chronic diseases such as MS. The discrepancy between DNAm-derived age and C-Age is obtained by regressing epigenetic age on chronological age and is referred to as epigenetic age acceleration (EAA).

Different generations of epigenetic clocks have been developed depending on their target and the algorithms used.

The first-generation clock, Horvath’s, observed that epigenetic age is accelerated in glial cells from brain tissue samples of MS participants compared with healthy controls [28].

“PhenoAge” is a second-generation epigenetic clock that was compared with three other measures of epigenetic change using blood samples from individuals diagnosed with multiple sclerosis (MS) [29]. This study demonstrated that these different measures capture distinct pathophysiological aspects of the disease and that people with MS exhibit significantly higher epigenetic age acceleration (EAA) based on DNAm PhenoAge in whole blood, independent of body mass index or smoking status. These findings are also observed in a pediatric population. A recent multicenter study [30] also observed an accelerated age in MS children in comparison with age-matched healthy controls using the PhenoAge clock.

Another second-generation epigenetic clock, the GrimAge clock, has been used to assess EAA in 583 MS patients in comparison to 643 non-MS controls. In this study the MS group exhibited an EAA increase of 5.1 years compared with controls [45].

These collective data suggest that according to epigenetic changes, aging is accelerated in MS and may contribute to pathology and disability severity.

### 3.3. Biomarker Composites

In contrast to TL or epigenetic clocks, multi-biomarker approaches have emerged as easily accessible measures of B-Age. These approaches capture the global effects of aging processes across multiple organ systems [46]. Multi-marker indices have proven successful in predicting mortality and estimating the risk of age-related diseases, including cardiovascular disease [46]. The 10-item U.S. National Health and Nutrition Examination Survey (NHANES) multi-marker index of biological age is a composite measure derived from each participant’s specific biological and clinical data, including the analysis of selected biomarkers such as creatinine, C-reactive protein, blood urea nitrogen, albumin, alkaline phosphatase, cholesterol, CMV IgG, hemoglobin A1c, Forced Expiratory Volume in 1 sec (FEV1), and blood pressure. This multi-marker index has been tested in 51 MS patients [31], finding that MS participants were biologically older than their age-matched controls.

## 4. MS Progression and Aging

Although clinical progression often manifests later in life and with longer disease duration, there is a growing body of evidence demonstrating that neurodegeneration and progression occur from disease onset [47]. This means that neurodegenerative mechanisms are active long before the occurrence of the clinical progressive course and gradually become more evident by growing older. From a clinical point of view, the main aspects that clinicians take into account when evaluating the prognostic risk of a given patient are radiological parameters and some biological and demographic baseline data [48]. They are useful, but not enough to properly classify patients and understand the heterogenic trajectories of progression. There are likely additional aspects that influence the disease progression phenomenon, in parallel with the passing of time. During the natural aging process, the immune system undergoes profound alterations in both composition and function. This phenomenon, known as immunosenescence, leads to a diminished adaptive immune response, greater susceptibility to infections, and increased production of non-organ-specific autoantibodies [49]. Senescent cells are broadly characterized by cell cycle arrest and resistance to apoptosis. Most senescent cells acquire a senescence-associated secretory phenotype (SASP), defined by the release of catabolic and pro-inflammatory factors [50]. Regardless of chronological age, cellular senescence can be triggered by a variety of stimuli—including telomere attrition, oncogenic signals, and cellular stressors such as oxidative or genotoxic damage and cytokine exposure—all of which contribute to SASP activation and the senescent phenotype [50]. Collectively, these changes promote systemic inflammation, impair tissue regeneration, and ultimately drive tissue degeneration. In MS patients, senescence has a double effect in the periphery and in the central nervous system, resulting in reduced adaptive immune response, reduced capacity of remyelination and tissue reparation, reduced efficacy of disease-modifying treatments and increased risk of treatment side effects such as infections [51,52].

## 5. Senomorphic and Senolytic Drugs in Multiple Sclerosis

Improved understanding of the relationship between aging and MS progression is important because targeting aging-related mechanisms is a potential therapeutic strategy for MS.

Senotherapy is an emerging field of research focused on developing therapeutic strategies that specifically target cellular senescence. There are currently more than 30 compounds that target senescence pathways [53]. Within this therapeutic group, we can find senomorphic and senolytic agents. Senomorphics are drugs that block or neutralize senescence-associated secretory phenotype (SASP) factors to reduce chronic inflammation and neurotoxicity.

Senomorphics can be classified into several subtypes according to their mechanisms of action, including mitochondrial antioxidants, Wnt/β-catenin inhibitors, JAK inhibitors, sirtuin modulators, mTOR inhibitors, and AMPK activators [54]. There are multiple studies in EAE, the experimental autoimmune encephalomyelitis mouse model of MS, that show that these compounds alleviated clinical disease and decreased demyelination [52,53,54,55]. In MS patients there are few early-phase clinical trials using senomorphics, with most of them having unreported data and some in progressive forms but without clear conclusions [55].

Another class of therapeutics commonly used to target senescence pathways are senolytics [56,57,58]. Senolytics are drugs that selectively initiate apoptosis in senescent cells by inhibiting the pro-survival mechanisms upregulated during senescence. These upregulated pathways were discovered by bioinformatics and transcriptomics of senescent cells when developing different generations of senolytic drugs [59].

While phase II clinical trials are underway in other neurodegenerative disorders (e.g., combinations such as dasatinib plus quercetin in Alzheimer disease) [60], MS-specific trials are lacking. Preclinical work suggests that senolytics can deplete senescent microglia, reduce inflammation, and improve neuronal survival in the EAE model. Navitoclax, a senolytic BCL2 inhibitor, showed an effect on EAE mice, decreasing motor symptom severity, improving visual acuity, promoting neuronal survival, and decreasing white matter inflammation [61].

Emerging in vitro and preclinical data in other diseases suggest that targeting cellular senescence may promote repair, remyelination, and neuroprotection and this could also work for MS and other demyelinating diseases. However, evidence of its effectiveness is still in its infancy, as is evidence of its potential toxicity. Dasatinib, a senolytic EMA approved drug [62] for treatment of chronic myeloid leukemia and lymphoblastic leukemia, has been related to the development of adverse effects such as cytopenias or pleural effusion. Other unmet needs for its use in MS include the lack of direct CNS biomarkers for senescence, which would help in its inclusion as an endpoint in clinical trials.

## 6. Lifestyle Habits and Biological Age

Because B-Age is partly modifiable, lifestyle represents a potential therapeutic target (Figure 1).

Among lifestyle factors, there is scientific evidence on how physical exercise, smoking, and obesity impact biological age markers in healthy people. The relationship between telomeres and physical exercise has been well documented. Endurance athletes exhibit higher telomerase activity and a slower rate of telomere attrition compared to inactive individuals [63]. Moreover, moderate levels of physical activity have been shown to attenuate telomere shortening in healthy populations compared to inactive populations [64]. A recent systematic review and meta-analysis suggested that the type of exercise is a relevant factor, reporting that high-intensity interval training (HIIT) exerts a more pronounced positive effect on telomere length than resistance or aerobic training in healthy individuals [65]. The beneficial impact of exercise has also been observed in other biological age (B-Age) markers, such as epigenetic clocks [66]. Smoking, conversely, is a well-established risk factor associated with conversion to the progressive phase of MS [67]. It enhances oxidative DNA damage, thereby accelerating telomere shortening. A meta-analysis of 30 studies confirmed the presence of significantly shorter telomeres in peripheral blood cells among ever-smokers compared to never-smokers, and among current smokers compared to former smokers, in individuals without other comorbidities [68]. In MS populations, studies have also reported epigenetic age acceleration in blood immune cells of smokers with MS [69].

Furthermore, obesity during childhood and adolescence represents a significant risk factor for MS, largely linked to unhealthy eating habits and a sedentary lifestyle.

A recent systematic review of 16 articles [70] found a negative association between childhood obesity and TL.

However, the interplay of physical activity, smoking, or obesity and B-Age among patients with MS has not been fully elucidated. Additional prospective research is needed to clearly define how lifestyle changes can slow down disease progression.

## 7. Conclusions and Final Remarks: Unveiling Future Directions

The influence of age and aging on the MS course has been highlighted since early epidemiological studies. Notably in recent years there has been growing evidence to suggest that the speed and way of aging could modulate the speed of MS progression, and that the relevance of biological age is growing in MS and other age-related diseases.

The measurement of biological age, however, faces several problems. Firstly, the techniques used are not easily accessible and there is low agreement between their results. Secondly, the use of drugs intended to reverse senescence is far from showing significant efficacy in patients with MS and other neurodegenerative diseases.

The speed of aging can be influenced, at least in part, by our lifestyle, which is a modifiable issue. Current approved therapies for MS do not stop progression, and the elderly MS population has increased in our centers. We need to generate evidence about the impact that a healthy lifestyle, an affordable “add-on therapy” for all patients, has on biological age, aging and hypothetically on disability progression in MS patients.

## Figures and Tables

**Figure 1 healthcare-13-02619-f001:**
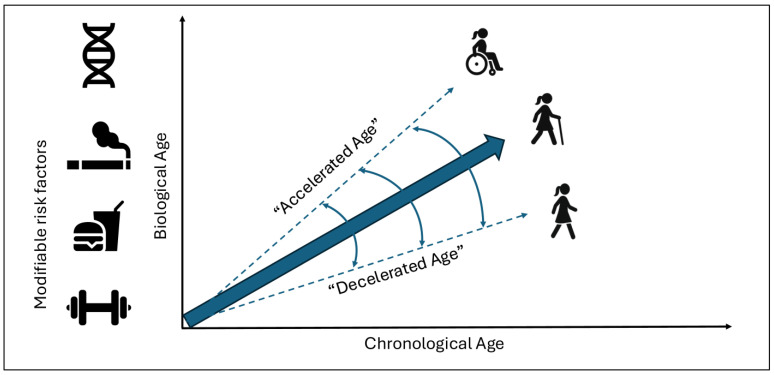
Relation between chronological age and biological age. Biological age is determined by genetic and modifiable factors, leading to accelerated or decelerated aging, hypothetically impacting the patient’s clinical trajectory.

**Table 1 healthcare-13-02619-t001:** Biological age biomarkers in multiple sclerosis (MS).

Biological Age Biomarker	Current Evidence in MS	References
Telomere length	‑ Systematic review: shorter leukocyte telomeres in MS patients associated with greater disability, lower brain volume, increased relapse rate, and faster conversion to progressive MS.	Bühring, 2021 [24]
	‑ MS patients (n = 59) vs. healthy controls (HC, n = 60): significantly shorter telomeres in the primary progressive MS (PPMS) group; no significant difference in other subtypes.	Guan, 2015 [25]
	‑ MS patients (n = 510): shorter telomere length associated with disability, independent of chronological age.	Krysko, 2019 [26]
	‑ MS patients (n = 138) vs. HC (n = 120): significantly shorter telomere length in MS group.	Habib, 2020 [27]
Epigenetic clocks	**First-generation clocks:**	
	‑ Significant increase in biological age in glial cells of MS patients compared to HC.	Kular, 2022 [28]
	**Second-generation clocks:**	
	‑ MS patients showed significantly higher epigenetic age acceleration (EAA) than HC, independent of body mass index and smoking.	Theodoropoulou, 2019 [29]
	‑ Pediatric MS patients (n = 125) had significantly higher biological age than age-matched HC (n = 154).	Goyne, 2025 [30]
Biomarker composites	‑ NHANES Biological Age Index: MS patients (n = 51) were biologically older than age-matched HC (n = 38).	Miner, 2023 [31]

Abbreviations: MS, multiple sclerosis; HC, healthy controls; PPMS, primary progressive multiple sclerosis; EAA, epigenetic age acceleration.

## Data Availability

No new data were created or analyzed in this study. Data sharing is not applicable to this article.

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
