# Peer review of "The Relevance of Chronological and Biological Aging in the Progression of Multiple Sclerosis"

_healthcare, 2025, doi:10.3390/healthcare13202619_

Round 1
Reviewer 1 Report
Comments and Suggestions for Authors
I read this manuscript with interest and found it to be beneficial.
-The Lifestyle section should be expanded to include specific exercise recommendations. Can be displayed, for example, in a clear table.
-in the introduction section there could be an even more detailed comparison of problematic biological vs. chronological age in other diseases.
Author Response
Dear reviewer 1,
Thank you very much for taking the time to review this manuscript. Please find the detailed responses below and the corresponding corrections in track changes in the re-submitted files.
Reviewer comment:
The Lifestyle section should be expanded to include specific exercise recommendations. Can be displayed, for example, in a clear table.
Response:
Thank you very much for point out this suggestion. Lifestyle section have been rewritten to clarify some aspects. However, I consider that the inclusion of a specific exercise recommendations goes beyond the aim of this article.
Reviewer comment:
In the introduction section there could be an even more detailed comparison of problematic biological vs. chronological age in other diseases.
Response:
Introduction have been rewritten a this concept have been more detailed.
Reviewer 2 Report
Comments and Suggestions for Authors
The presented work demonstrates the relevance of chronological and biological age in MS.
- Figure 1. Please indicate that accelerated/decelerated age correlates with increased or decreased disability in MS
- Figure 2, is a general disease progression figure for MS. How do chronological and biological age contribute to the disease progression?
- Review paper should not look like a textbook; Please address a more in-depth comparison between chronological and biological age in MS, especially what would be beneficial for clinical diagnosis and treatment.
- Please provide a detailed table summarizing clinical/experimental evidence for the relationship between chronological/biological age and MS.
Author Response
Dear Reviewer 2,
Thank you very much for taking the time to review this manuscript and for your comments, which I sincerely believe have helped improve the quality of our article.
Please find the detailed responses below and the corresponding corrections in track changes in the re-submitted files.
Comment 1:
- Figure 1. Please indicate that accelerated/decelerated age correlates with increased or decreased disability in MS
Response:
We have modified Figure 1 in order to better reflect the concept that accelerated biological age shortens the time to disability acquisition.
Comment 2:
2. Figure 2, is a general disease progression figure for MS. How do chronological and biological age contribute to the disease progression?
Response:
We agree with your comment and we have eliminated figure 2 and include instead a table summarizing the current evidence in biological age biomarkers in MS as you suggest in your comment 4.
Comment 3:
Review paper should not look like a textbook; Please address a more in-depth comparison between chronological and biological age in MS, especially what would be beneficial for clinical diagnosis and treatment.
Response:
That section has been revised and deepened for a better explanation.
Comment 4:
Please provide a detailed table summarizing clinical/experimental evidence for the relationship between chronological/biological age and MS
Response:
Thank you for this suggestion that clearly has improved our article. Table has been included.
Reviewer 3 Report
Comments and Suggestions for Authors
To establish a strong foundation for the concept of biological age, more technical detail and robust measurement methods are needed. This area requires a clearer definition and identification of precise biomarkers to support its clinical relevance.
Additional references are needed to substantiate the link between biological age and disability progression. Notably, disease progression often occurs in a different age group—typically after ~15 years from RRMS onset when PMS begins. The manuscript should clarify the transition phase between RRMS and PMS.
The mention of senolytics as a therapeutic avenue is interesting, but given the limited evidence in MS, it should be discussed more cautiously.
The potential impact of lifestyle on biological age is promising but remains speculative; this should be acknowledged appropriately.
Lastly, the manuscript would benefit from minor language edits and improved structure for better clarity and flow.
Comments on the Quality of English LanguageTo establish a strong foundation for the concept of biological age, more technical detail and robust measurement methods are needed. This area requires a clearer definition and identification of precise biomarkers to support its clinical relevance.
Additional references are needed to substantiate the link between biological age and disability progression. Notably, disease progression often occurs in a different age group—typically after ~15 years from RRMS onset when PMS begins. The manuscript should clarify the transition phase between RRMS and PMS.
The mention of senolytics as a therapeutic avenue is interesting, but given the limited evidence in MS, it should be discussed more cautiously.
The potential impact of lifestyle on biological age is promising but remains speculative; this should be acknowledged appropriately.
Lastly, the manuscript would benefit from minor language edits and improved structure for better clarity and flow.
Author Response
Dear reviewer 3,
Thank you very much for taking the time to review this manuscript. Please find the detailed responses below and the corresponding corrections in track changes in the re-submitted files.
Comment 1: To establish a strong foundation for the concept of biological age, more technical detail and robust measurement methods are needed. This area requires a clearer definition and identification of precise biomarkers to support its clinical relevance.
Response:
Thank you very much for this comment. Completely agree that biological age is a concept "in development" that requires a more detailed explanation. The initial section has been rewritten to provide a better definition.
Also for a better explanation of this concept we have modified Figure 1 and we have included a table with the more relevant evidences of biological age biomarkers findings in MS pathology.
Comment 2: Additional references are needed to substantiate the link between biological age and disability progression. Notably, disease progression often occurs in a different age group—typically after ~15 years from RRMS onset when PMS begins. The manuscript should clarify the transition phase between RRMS and PMS
Response:
We have rewritten a more detailed section about the impact of the chronological age on MS progression
Comment 3: The mention of senolytics as a therapeutic avenue is interesting, but given the limited evidence in MS, it should be discussed more cautiously.
Response:
Thank you for point out this. We have rewritten the section with particular care to clarify that the evidence for this therapy is currently limited.
Comment 4: The potential impact of lifestyle on biological age is promising but remains speculative; this should be acknowledged appropriately.
Response: Completely agree. We have carefully revised the manuscript to clarify this concept.
Comment 5: Lastly, the manuscript would benefit from minor language edits and improved structure for better clarity and flow.
Response:
The manuscript has been fully revised for structural and grammatical improvement.
Reviewer 4 Report
Comments and Suggestions for Authors
This is a well-structured and timely review that addresses the important distinction between chronological and biological aging in the context of multiple sclerosis (MS). The manuscript integrates emerging concepts such as epigenetic clocks and senescence-targeting therapies, and highlights how B-Age might refine prognostic assessment and therapeutic strategies in MS. However, there are several problems and suggestions as follows:
1.Title
Here is a suggestion title(optional): The Relevance of Chronological and Biological Aging in the Progression of Multiple Sclerosis
2. Introduction
This section should effectively introduce the distinction between chronological and biological age. Good rationale for the topic and clarify the thesis of the paper earlier: e.g., “B-Age may be a better predictor than C-Age in MS progression.”
3. To make the manuscript easier to understand to even non-specialist readers, there are several words need to define as it is first mentioned, such as "GrimAge clock”, “senolytic” and “senomorphic”, etc.
4. Suggest the author to replace "immune senesce" (Line 185) with "immunosenescence”.
5. Suggest a summary table comparing TL, epigenetic clocks, and biomarker composites (with strengths, weaknesses, and current evidence in MS).
6. Lifestyle and Biological Aging
About this section, it’s better to provide some information to confirm the relationship between lifestyle findings specifically to MS (not just general population). Furthermore, suggest the author consider discussing feasibility of interventions in MS patients (e.g., fatigue limiting exercise).
Author Response
Dear reviewer 4,
Thank you very much for taking the time to review this manuscript. Please find the detailed responses below and the corresponding corrections in track changes in the re-submitted files.
Comment 1:
Here is a suggestion title(optional): The Relevance of Chronological and Biological Aging in the Progression of Multiple Sclerosis
Response:
I consider that your title suggestion better reflects the content of the article, so we have accepted it.
Comment 2:
This section should effectively introduce the distinction between chronological and biological age. Good rationale for the topic and clarify the thesis of the paper earlier: e.g., “B-Age may be a better predictor than C-Age in MS progression.
Response:
We have rewritten this section to a better explanation of this concept.
Comment 3:
To make the manuscript easier to understand to even non-specialist readers, there are several words need to define as it is first mentioned, such as "GrimAge clock”, “senolytic” and “senomorphic”, etc.
Response:
Thank you very much for point out this. We have included a more detailed descriptions of this terms.
Comment 4:
Suggest the author to replace "immune senesce" (Line 185) with "immunosenescence”.
Response:
Thank you for point this mistake. Change have been made.
Comment 5:
Suggest a summary table comparing TL, epigenetic clocks, and biomarker composites (with strengths, weaknesses, and current evidence in MS)
Response:
A new table that summarize current evidence of biological aging biomarkers in MS have been included
Comment 6:
Lifestyle and Biological Aging. About this section, it’s better to provide some information to confirm the relationship between lifestyle findings specifically to MS (not just general population). Furthermore, suggest the author consider discussing feasibility of interventions in MS patients (e.g., fatigue limiting exercise)
Response:
Thank you very much for this comment. We have included some new data about biological age measured by epigenetic clocks in smoking MS patients.
Round 2
Reviewer 3 Report
Comments and Suggestions for Authors
The manuscript has been thoroughly revised, and the authors have demonstrated a clear understanding of their research and perspectives. It is recommended for acceptance by the journal.